# Health Risk of Heavy Metals Related to Consumption of Vegetables in Areas of Industrial Impact in the Republic of Kazakhstan—Case Study for Oskemen

**DOI:** 10.3390/ijerph20010275

**Published:** 2022-12-24

**Authors:** Laura Boluspayeva, Monika Jakubus, Waldemar Spychalski, Akhan Abzhalelov, Yertas Bitmanov

**Affiliations:** 1Department of Management and Engineering in the Field of Environmental Protection, L. N. Gumilyov Eurasian National University, Astana 010000, Kazakhstan; 2Department of Soil Science and Microbiology, Poznan University of Life Sciences, 60-637 Poznań, Poland

**Keywords:** bioconcentration factors (BCF), daily intake of metal (DIM), health risk index (HRI), industrial and urban zones, allotment gardens

## Abstract

Among various heavy metal sources the metallurgic industry is the most threatening because emitted metals presented are the chemical forms in which metals are found in soil are more bioavailable and thus very easily are introduced into the environment and spread in both soils and plants. In this study such a situation is presented and the potential negative effect of emitted metals on soil and vegetables is estimated. Therefore, the following indicators were used: bioconcentration factors calculated for the total amount of metals (BCF) as well as daily intake of metal (DIM) and health risk index (HRI). Analyzed soils and vegetables originated from allotment gardens located at different distances from local industrial plants. The greatest amounts of metals in investigated materials (soils and plants) were found for the industrial zone and the lowest for samples representing the suburban zone. Among the analyzed metals Zn showed the highest (223.94–2645.13 mg·kg^−1^ for soils and 9.14–49.28 mg·kg^−1^ for plants), and Cd the lowest levels (1.77–15.2 mg·kg^−1^ for soils and 0.05–0.46 mg·kg^−1^ for plants). Regardless of the metal, the lowest BCF values were calculated for plants from the industrial zone and the highest from the urban site. Generally, BCF values calculated for vegetables were low and comparable for carrots, tomatoes, and cabbage. BCF values obtained for beetroots were higher in comparison to other vegetables. Regardless of plants, DIM values for Cd and Pb were low and comparable. DIM values for Cu and Zn were higher, but simultaneously strongly differentiated depending on the analyzed vegetables. A similar tendency was found in the case of HRI. The highest values were recorded for Cu and Zn in tomatoes. Regardless of the individual metals, the calculated values for DIM and HRI indices increased in the following sequence: beetroot < cabbage < carrot < tomato. The Zn and Cu contents in the studied types of vegetables do not exceed the maximum permissible levels recommended by WHO/FAO. In contrast, Pb concentrations were higher than the imposed standards in all the analyzed vegetable samples. On the basis of obtained DIM and HRI indices, consumption of vegetables cultivated in industrial areas should be restricted due to health risks related to heavy metals contained in plants.

## 1. Introduction

According to the United States Environmental Protection Agency (EPA) compilation, seven heavy metals, Pb, Cr, Zn, Cu, Cd, Hg, Ni, and a metalloid such as As, are listed to be the most widespread heavy metals in the environment [1]. Heavy metals constitute a group of elements characterized by high density and high toxicity even at low concentrations in the environment, so taking under consideration metal toxicity to living organisms Zwolak et al. [2] proposed the following order: Hg > Cu > Zn > Ni > Pb > Cd > Cr > Sn > Fe > Mn > Al. The group of heavy metals is very differentiated because there are both very toxic, non-essential metals (Pb, Cd, Hg, Al) and essential micronutrients as Cu, Ni, and Zn. The latter metals play an important role in the metabolic pathways during plant growth and development when available in required concentrations, but not higher. Heavy metals continue to be released into the environment posing a threat to the health of people exposed to them both through inhalation and food and these hazardous pollutants originate from many sources, first of all, mining, smelting activity, steel and iron industry, chemical industry, and transport [3,4,5]. The emitted metals penetrate into environments where soil is their primary reservoir. Therefore, this environment component plays an essential role in global metal cycles. It is of particular importance in the case of agricultural and horticultural soils, where heavy metals may be transferred from the soil to the cultivated plants and accumulated in their tissues. Contamination through different food chains is the major pathway of heavy metal exposure for humans. Moreover, humans are the last link in the food chain so they are particularly exposed to the negative effects of heavy metals [4,6,7]. Thus, direct consumption of fruits and vegetables is mainly responsible for the potential contamination of human organisms. Moreover, it is now widely recognised that heavy metals taken up by plants constitute a predominant source of their accumulation in food [8]. In view of the above, special attention is focused on plants cultivated in close surroundings of metallurgic plants as well as mines and smelting works. Very often the concentration of heavy metals in plants is not assessed or controlled while at the same time posing a significant risk of contamination of the human diet, particularly, when inhabitants cultivate vegetables and other crop plants for their own needs in family allotment gardens. Family allotment gardens are very popular and play important functions both for cities and their inhabitants. Unfortunately, such gardens were established in relatively unattractive sites, near industrial plants, railways, and car transport routes [9]. As a consequence, the obtained plant products may be contaminated with heavy metals emitted from nearby sources. Consumption of such fruits or vegetables can greatly affect human health. Gupta et al. [4] indicated that dietary intake of metals through contaminated vegetables may cause various chronic diseases. Sandeep et al. [5] discussed this issue in detail, underlining human health implications of individual heavy metals. The negative impact of heavy metals contained in plant products on the human will directly depend on the quantity and quality of consumed products, in this case heavy metal amounts. Such a potential risk should be assessed in several areas, taking into account the effectiveness of metal bioconcentration in organs and tissues of consumed vegetables and the daily intake of metals through consumptions of various plant products. Based on the amount of metals in the soil and in plants the transfer of heavy metals from soil to plants should be assessed, with bioconcentration factors (BCF) being useful indices in this respect [10,11]. Furthermore, the probable human risk associated with heavy metals should also be evaluated. Various indices and parameters are used to assess the human health risk posed by heavy metals through vegetable consumption. Gupta et al. [4], Nag et al. [6] and Kumar et al. [12] listed such indices including the target, hazard quotient, daily dietary intake of metals, hazard index, daily intake of metals and health risk index. A literature review shows that the most popular indices are daily intake of metals (DIM) and health risk index (HRI) [9,11,13,14,15,16]. Daily intake of metals can help assess the relative phyto-availability of metals and is proportional to metal concentrations in the edible parts of vegetables as well as the consumed amounts of those vegetables. The health risk index underlines the significance of DIM in relation to the reference dose (RfD) of each metal. The value of HRI < 1 assumes no risk of a non-carcinogenic effect and the exposed population is said to be safe. HRI >1 is assumed to have potential significant non-carcinogenic effects.

In this study, the above indices were used for the first time to assess the potential threat to soil and plants posed by the metallurgical industry and its heavy metals emissions in the largest industrial city (Oskemen) in the Republic of Kazakhstan. The total area of the city is 230 km^2^ and it is inhabited by 300,000 people. There are over 400 plants in the city, which together emit almost 70,000 Mg of hazardous substances, including mainly copper, zinc, lead, and cadmium compounds. The dominant reason for undertaking this research was to draw the attention of the local community, especially decision-makers, to the problem of the presence of heavy metals in the environment with its dangerous and direct impact on human health as a result of both cultivation and consumption of vegetables potentially loaded with metals.

Therefore, the present study is aimed: 1. to determine heavy metal amounts in soils and edible parts of selected vegetables: carrot, cabbage, tomato, and beetroot, 2. to assess the potential intensity of heavy metal concentrations in edible parts of vegetables as the bioconcentration factor (BCF), and 3. to investigate the potential health hazard associated with both daily intake of metals (DIM) and health risk index (HRI). 

## 2. Materials and Methods

### 2.1. Study Area

The study was conducted in four separate zones (I—central urban zone, II—suburban zone, III—industrial zone, IV—north-eastern industrial zone) in the city of Oskemen (Figure 1). It is the largest industrial city in the northeast of the Republic of Kazakhstan, where the mining industry and metallurgy are very well developed. The applied division of the city into zones was dictated by the distribution of industry and the average population density. The authors took into account the intensity of the industrial load, the distance from emission sources and the number of residential buildings. The urban area (I zone) includes a significant part of the residential areas of the regional center. The suburban southern area (II zone) includes: a silk fabric factory, the Left Coast, Ablaketka, and residential areas in neighborhood of the Okemen hydroelectric power station. The northern industrial zone (III zone) is represented by the industrial sites of UKMC Kazzinc LLP, Ulba Metallurgical Plant JSC, AES JSC Oskemen CHPP and the territories surrounded to them. On the north-eastern industrial area (IV zone) the JSC Oskemen Titanium-magnesium plant and Sogrinsk thermal power station are located. According to WRB [17], the soils of the study area are primarily represented by *Haplic Chernozems, Gleyic Chernozems* and *Luvic Chernozems.* The soil samples were taken from a depth of 0–20 cm from allotment gardens located in four zones of the study area (Figure 1). Each soil sample of approximately 0.5 kg collected in selected sites consisted of 15 individual punctures with a soil sampler. The collected samples were dried to an air-dry state, then ground in a porcelain mortar and sieved through a nylon sieve with a mesh size of 2 mm. According to Boluspaeva et al. [18], soils in the study area are characterized by a sandy silt or loam sand texture, which was shown by the percentage of clay ranging from 2.9 to 6.1% (Table 1). The pH of the analysed soils ranged from neutral to basic, and its values expressed in pH units ranged from 6.42 to 7.70. The cation exchange capacity (CEC) was relatively high and reached 8.1–25.3 cmol (+)·kg^−1^. The total carbon (TC) content ranged from 0.98 to 5.55%, but most often exceeded the value of 2% (Table 1). The mean amount of total nitrogen (TN) was 0.20% in the range from 0.09 to 0.36%. The C:N values oscillated around 12:1 (Table 1). 

The following vegetables, carrot, tomato, cabbage and red beetroot, were collected in the same locations as soil samples. All vegetable samples were washed with distilled water, crushed and dried at 60 °C. After drying they were homogenized in a mortar.

### 2.2. Methods

Soil texture was determined using a laser particle size analyzer Mastersizer 2000 with a Hydro MU dispersion unit (Malvern Panalytical, Malvern, UK). The chemical composition of the soils was analyzed in the <2 mm fraction. The soil pH was measured potentiometrically in 0.01 M CaCl_2_ solution with a soil/extractant ratio of 1:5. Sorption capacity was determined in an extract of 1 molar ammonium acetate with a pH of 7.0. The total carbon (TC) and total nitrogen (TN) were analyzed using a VarioMax elemental analyzer (Elementar Analysensysteme GmbH, Langenselbold, Germany). 

The total metal contents in the soil were determined spectrophotometrically according to the ISO procedure [19]. The plant material (aboveground parts) was dried at 60 °C, ground and ashed in a furnace at 450 °C for 6 h. The ash was dissolved in 5 mL of 6 mol∙dm^3^ HCl and diluted to a constant volume with distilled water [20]. The obtained soil and plant extracts were analyzed to measure the metal contents by atomic absorption spectrophotometry (ASA) in a Varian Spectra AA220 FS apparatus. The method employed to determine the examined elements in all the extracts was atomic absorption spectrometry with atomisation in a flame of acetylene/air. To evaluate the validation of the obtained laboratory test results, the certified reference material RTH 907 was used. Recovery of the analyzed elements was achieved at the level of 94 to 98%.

To express the plants’ ability to take up and transport the most mobile and bioavailable amounts of heavy metals and to assess the actual site contamination with heavy metals the specified factor was calculated. The metal contents in the soil and vegetables were used to calculate the bioconcentration factors for individual metals. On the basis of heavy metal total contents in the soil and in cultivated plants the bioconcentration factors were calculated as follows [11]:$$\mathrm{BCF}=\frac{\mathrm{metal}\;\mathrm{in}\;\mathrm{plant}\;}{\mathrm{metal}\;\mathrm{in}\;\mathrm{soil}}$$

Additionally, the daily intake of metal (DIM) and the health risk index (HRI) were calculated. For the daily intake of metal (DIM) the following equation was used [16]: $$\mathrm{DIM}\;[\mu g\cdot {{\mathrm{kg}}^{-}}^{1}]=\frac{\mathrm{metal}\;\mathrm{in}\;\mathrm{plant}\cdot \mathrm{average}\;\mathrm{daily}\;\mathrm{consumption}*\;}{\mathrm{body}\;\mathrm{weight}**}$$

*—based on WHO data [21] adults in Europe have an average daily consumption of carrots amounting to 2.8, tomatoes 44.1, cabbage 5.0 and beetroot 0.5 g/person 

**—based on world statistics the average body weight of adults is 62.6 kg (average weight of men and women) 

The health risk index (HRI) was calculated using the following equation given by Jan et al. [22], cited after [14]: $$\mathrm{HRI}=\frac{\mathrm{DIM}\;}{\mathrm{RfD}*}$$

*—RfD—reference oral dose; the values for Cu, Zn, Pb, and Cd amounted to 0.04, 0.30, 0.004 and 0.001 (mg/kg/day), respectively (values given according to US-EPA [23]).

### 2.3. Statistical Analysis

The data presented in the paper are means of five replications. The data were compiled applying one-way ANOVA. Each of the parameters for individual vegetables was tested independently using the F-test at the significance level α = 0.95. The assumed null hypothesis was that the mean values of the examined parameter are equal for each of the analyzed vegetables against the alternative hypothesis that not all the means are equal. As a result of the rejection of the null hypothesis the least significant differences were calculated using the Tukey test at the significance level α = 0.05. Tukey’s analysis was performed to distinguish homogeneous groups among the analyzed parameters individually for vegetables. The data were analyzed using the STATOBL software working in the Windows environment. The obtained results are visualized in the form of boxplots. In a standardized way of displaying the distribution of data the minimal value, first quartile (Q1), median, third quartile (Q3), and maximal value are given.

## 3. Results

The amounts of the analyzed heavy metals in the soils from individual studied zones decreased in the following sequence: III > IV > I > II. The quantitative differences between the minimum and maximum amounts were 5.0 times (Cu), 8 times (Cd), 12 times (Zn) and 16 times (Pb), which was statistically confirmed (Table 2). Similar relationships can be noted with regard to the contents of heavy metals in the tested vegetables (Table 2). The highest quantities of heavy metals were present in plants grown in the industrial zone and the lowest in plants grown in the suburbs. Analysed soils showed the highest amounts of Zn and the lowest of Cd and this directly influences the same relations in individual vegetables. It should be emphasized that the amounts of Pb and Cd in tomatoes grown in different zones did not differ significantly. It is worth noting that generally the amounts of metals found in vegetables cultivated in zones I and II did not differ statistically. A similar situation can be indicated in the case of plants grown in zones III and IV. Regardless of the zone, beetroots accumulated the highest, and tomatoes accumulated the lowest amounts of metals. As it was marked in the case of soils, also in relation to the tested vegetables the differences between the extreme values were determined, although they were not as spectacular as for soils. Regardless of the vegetable, the differences in the amount of Cu were 2–4 times, for Zn 1.5–2.0 times, for Pb 3.0 times, and for Cd 2.5–4.0 times (Table 2). 

On the basis of the metal amounts in the soil and the plants, the BCF value was calculated. BCF characterizes the theoretical transfer and concentration of a metal from the soil to the plant. Regardless of the cultivated plant and the analyzed metal, the lowest BCF values were determined in plants grown in the industrial zone (III) and the highest in the central urban zone (I). Independently of the analysed vegetables, the BCF values increased in the following sequence: Cu > Zn > Cd > Pb. Simultaneously one can notice that BCF values for plants, regardless of the metals and their amounts, decrease as follows: tomato < cabbage < carrot < beetroot. As can be seen from Figure 2, for carrots the BCF average values for Cu ranged from 0.07 to 0.15, while for Zn it was from 0.01 to 0.09. Similar levels of values were recorded for BCF Cd (0.02–0.08). The mean values of BCF for Cu, Cd, and Zn found for carrots cultivated in the industrial zone significantly differed from those recorded in the other zones. No influence of heavy metals emitted by the local industry was found in relation to BCF for Pb (0.01–0.02), because the differences between the obtained values were non-significant. 

Also, the industry had a weak influence on the differentiation of BCF mean values for the tested metals in tomato, because the differences were non-significant in the case of Cu (0.05–0.09) and Pb (0.007–0.01) (Figure 3). The values of BCF for Zn and Cd calculated for tomatoes cultivated in zones I and II as well as III and IV did not differ statistically. The BCF values for Zn ranged from 0.01 to 0.06 and for Cd from 0.01 to 0.05. 

For cabbage, similarly as for carrots or tomatoes, also low BCF values (0.008–0.1) were determined and it was independent of the metal. Simultaneously the data calculated for BCF Pb as well as for Cd were comparable between the individual zones where the discussed vegetable was grown (Figure 4). The average values of BCF of Cu for cabbage from zone I (on average 0.1) were the highest and statistically different from the other calculated values (0.05–0.07). BCF values for Zn in cabbages grown in zone III (on average 0.01) were significantly lower compared to the values found for the discussed vegetable from the other zones (0.04–0.05) (Figure 4). 

Of all the cultivated vegetables, the highest BCF mean values were found for beetroot (Figure 5). The direct influence of industry in zone III was marked only in relation to Cu and Zn, for which the BCF values were statistically lower (on average 0.09 for Cu and 0.02 for Zn) compared to the other values (0.16–0.23 for Cu and 0.09–0.11 for Zn). The BCF values for Pb and Cd were similar (0.01–0.03 for Pb and 0.03–0.06) and did not differ statistically (Figure 5).

The intensity of the industry’s impact on the quality of vegetables should not only be assessed on the basis of metal concentrations in their edible parts, but the potential impact of heavy metals included in vegetables on human health should be also considered. Therefore, the influence of the anthropogenic factor on changes in DIM and HRI values was analyzed. Regardless of the tested vegetable and metal, the highest values of DIM and HRI were determined in plants grown in the industrial zone (III) and the lowest in the suburban zone (II). Moreover, the DIM values depended on the metal and increased in the following sequence: Cd < Pb < Zn < Cu. Also, in the case of HRI values, these values decrease as follows: Cu > Pb > Cd > Zn. Regardless of the analyzed metals, DIM values increase in vegetables in the following sequence: beetroot < cabbage < carrot < tomato. In the case of HRI values, one may notice a slightly different order: beetroot < cabbage < tomato < carrot. For carrot the DIM values for Cu ranged on average from 0.41 μg∙kg^−1^ to 1.02 μg∙kg^−1^, for Zn it was from 0.87 μg∙kg^−1^ to 1.43 μg∙kg^−1^, while the values of the discussed parameter in both cases did not differ statistically for plants from zones I and II and for zones III and IV. In turn, DIM for Pb and Cd in carrot showed lower mean values ranging from 0.04 μg∙kg^−1^ to 0.12 μg∙kg^−1^ (Pb) and from 0.005 μg∙kg^−1^ to 0.01 μg∙kg^−1^ (Cd) (Figure 6). 

As shown in Figure 7, the HRI values determined for carrots indicated a significant differentiation depending on the analyzed metal. Regardless of the above, the significantly lower HRI values were found for carrots cultivated in zone II, as they were on average 10.62, 10.0, 5.19 and 2.88 for Cu, Pb, Cd, nd Zn, respectively. Carrots grown in the industrial zone had the highest HRI values. At the same time, it should be noted that the HRI values determined for zones III and IV did not differ significantly, similarly to those for zones I and II (Figure 7).

As shown in Figure 8, the influence of industry emitted heavy metals on DIM values in tomato was strongly marked. The mean values of DIM for Cu and Zn determined for this vegetable grown in the industrial zone (III) differed significantly from the other data. Regardless of the above, the lowest values were determined for tomatoes representing zone II (3.76 μg∙kg^−1^ for Cu; 7.21 μg∙kg^−1^ for Zn; 0.5 μg∙kg^−1^ for Pb, and 0.04 μg∙kg^−1^ for Cd), and the highest for those from zone III (11.88 μg∙kg^−1^ for Cu; 18.08 μg∙kg^−1^ for Zn; 1.2 μg∙kg^−1^ for Pb 0.12 μg∙kg^−1^ for Cd) (Figure 8). It is worth noticing that the DIM values calculated for Pb as well as Cd did not differ significantly. 

A similar relationship can be found in relation to HRI mean values in tomato (Figure 9). The calculated HRI values for this plant grown in zone III significantly differed from those in the other zones of the study area, being the highest (Cu: 296.94; Zn: 60.26; Pb; 306.46; Cd: 122.58). The lowest values of HRI were found in tomatoes representing the suburban zone (Cu: 94.05; Zn: 24.05; Pb: 119.87; Cd: 38.04) (Figure 9). Generally, the HRI values obtained for plants representing zones I, II, and IV did not differ significantly. 

As shown by the data in Figure 10, the DIM values of the analyzed metals for cabbage were significantly higher (4.0 times) for vegetables representing the industrial zone (III) compared to values calculated for plants cultivated in the other zones of the study area. The lowest values were recorded for cabbage from zone II: Cu—0.32 μg∙kg^−1^; Zn—0.73 μg∙kg^−1^; Pb—0.04 μg∙kg^−1^; and Cd—0.006 μg∙kg^−1^. Similar relationships can be found in the case of HRI values, which for Cu ranged from 8.11 to 33.51, for Zn from 2.43 to 7.19, for Pb from 10.8 to 35.5, and for Cd from 5.91 to 22.52 (Figure 11). The highest values were obtained for plants cultivated in the industrial area (III zone) and the lowest for the suburban area (II zone). Generally, the data calculated for vegetables from zones I; II, and III did not differ from one another. 

Among all the analyzed vegetables the lowest DIM and HRI values were determined for beetroot (Figure 12 and Figure 13), which was a phenomenon characteristic of all the tested metals. Nevertheless, the impact of industry in zone III was noted, because the plants grown there had the highest DIM values (2–4.0 times) in relation to the values found for beetroots from zones I and II. The DIM mean values for Cu ranged from 0.1 μg∙kg^−1^ to 0.3 μg∙kg^−1^, for Zn from 0.17 μg∙kg^−1^ to 0.39 μg∙kg^−1^, for Pb from 0.009 μg∙kg^−1^ to 0.03 μg∙kg^−1^, and for Cd from 0.001 μg∙kg^−1^ to 0.004 μg∙kg^−1^ (Figure 12). Data calculated for Cu, Zn, and Pb for plants representing zones I and II as well as III and IV were comparable, so the differences were non-significant. A slightly different situation can be noticed in DIM values for Cd, because a significant difference was found only for plants from zones I and II (Figure 12). According to the data in Figure 13, the HRI average values for beetroots ranged from 2.52 to 7.57 (Cu), 0.57 to 1.31 (Zn), 2.4 to 6.4 (Pb), and 0.94 to 3.64 (Cd). Regardless of the analyzed metal, the maximum values for beetroots grown in zone III significantly differed from those minimum HRI values found for plants representing zone II. Additionally, it should be stated that the HRI values calculated for beetroots grown in zones I and II as well as III and IV were comparable (Figure 13).

## 4. Discussion

The issue of horticulture in industrial cities can be controversial due to the potential contamination of crops with emissions from industrial plants. On the other hand, allotment gardens have a long tradition and are very popular in cities. Allotment gardens are a major form of urban agriculture in the world that act as an important feature of urban food security and sustainable urban food systems [24,25]. However, simultaneously such gardens are often exposed to several emission sources from the surrounding industry, which can lead to contamination of cultivated plants. Taking under consideration the possible pollution of soils and vegetables with heavy metals, this issue must be considered in terms of ecological risks and human health hazards. It is because soil acts as a reservoir of accumulated toxic metals, which are transferred to vegetables and enter the human food chain causing non-carcinogenic and carcinogenic health hazards due to consumption of contaminated vegetables [26]. Awareness of these dependencies is growing, although not at the same rate all over the world, as evidenced by the low interest in this subject in Kazakhstan. The diagnosis and analysis of this issue was carried out in a large industrial center—the city of Oskemen. According to Woszczyk et al. [27], the area of the city is exposed to the impact of polymetallic pollution and the contents of Cd, Cu, Pb, and Zn exceed values of the geochemical background for each element from moderate to extreme. According to Boluspaeva et al. [18], the average concentration of heavy metals in the soils of Oskemen exceeded the background level of studied metals in soils in this region (Cu—21.4 mg·kg^−1^, Zn—67.4 mg·kg^−1^, Pb—17.8 mg·kg^−1^, Cd—0.2 mg·kg^−1^) from 2 to 13 times. This study confirms the information cited above, because the average content of the analyzed metals varies widely in soils. In comparison to the literature data [28,29] presented for other industrial areas the amounts of heavy metals found in soils of Oskemen were higher. Taking under consideration the fact that there are more than 30,000 summer cottages and private houses located in the city and these areas are mainly intended by the residents for growing vegetables, it makes sense to assess the metal contents in soils according to hygienic standards. To assess the degree of soil pollution in Kazakhstan, an indicator is used such as the multiplicity of exceeding the maximum permissible level of chemicals. According to the approved order of the Ministry of Health of the Republic of Kazakhstan [30], the multiplicity of exceeding the limit level <1 means that the soils are clean, the multiplicity of exceeding the permissible level from 1 to 10 corresponds to heavily polluted soils. The obtained average values of the analyzed metals in soils in all the studied zones exceeded the maximum permissible level (Cu—33 mg·kg^−1^ and for Pb—32 mg·kg^−1^) adopted in Kazakhstan [31] by 2 to 11 times, which corresponds to heavily polluted soils. The contents of Cu, Pb, Zn, and Cd in zones I and II of the city are below the levels recommended by the Directive [32], where the maximum allowable limit for the content of metals in soil is as follows: for Cu 300 mg·kg^−1^, for Zn—300 mg·kg^−1^, for Pb 140 mg·kg^−1^, and for Cd 3 mg·kg^−1^. The contents of all the analyzed metals found in the industrial zone (III) were 3- to 9-fold greater in comparison to the recommended permissible levels given by Directive [32]. It should be noted that the average contents of heavy metals in the soils decreased noticeably depending on the distance from the emission source. Thus, the amounts of metals in soils from the industrial area (zone III) were from 2 to 12 times higher than those in soils from zone II (suburban area). The relatively low metal contents in soils of the suburban zone can be related not only to the distance from the emission source (13,000 m from the source of emissions, see Table 1), but also to the opposite direction of the wind rose. A comparison of the results of this study with the literature data [33,34] shows that the contents of Cd, Pb, Zn, and Cu determined for the soils of industrial areas were more than 10 times higher. On the other hand, Chowdhury et al. [35] found Cd in the range of 3.22–18.79 mg·kg^−1^ in soils and these values are quite comparable to those obtained in this study in soils from the industrial area (6.33–15.2 mg·kg^−1^). 

It should be noted that high heavy metal contents in the soils were correlated with those in plant products. The maximum contents of heavy metals were found in vegetables grown in zone III, and the minimum in zone II. These data are consistent with studies by Haque, et al. [36] or Zvolak [2], who noted significant heavy metal contamination in vegetables grown near industrial plants. Also, Zvolak [2], concluded that leaf and root crops are not suitable for growing in contaminated soils due to the high storage capacity of heavy metals. A significant risk of heavy metal-contaminated vegetables grown near industrial areas was confirmed by studies of Gupta et al. [4], Sun and Chen [37], Murray et al. [38], Sung and Park [39]. The range of heavy metal contents in vegetables grown on the soils of Oskemen vary greatly—from close to natural up to exceeding the hygienic standard proposed by FAO/WHO [40]. The safe limits for heavy metals in foodstuffs specified by the above-mentioned document are 73.0 mg·kg^−1^ for Cu, 99.0 mg·kg^−1^ for Zn, 0.3 mg·kg^−1^ for Pb, and 0.2 mg·kg^−1^ for Cd. When comparing these limits with the data presented in this study, it should be emphasized that the Cu and Zn amounts in the tested vegetables did not exceed the permissible level. At the same time, the contents of Cu and Zn in tomatoes were higher in comparison to results obtained by Haque et al. [36] or Chowdhury et al. [35]. Haque et al. [36] compared the Cu, Zn, Pb, and Cd contents in vegetables grown in industrial and non-industrial areas and found that significant quantitative differences clearly indicate the negative accumulation of metals in plants grown in industrial areas. These observations are confirmed in the presented paper. The outcomes from this study showed that Pb and Cd contents exceeded permissible thresholds [40] in vegetables grown in zones III and IV (with exception for Cd in tomato). The highest contents of Pb and Cd were recorded in beetroots and regardless of the cultivation site, the values were 2–11 times higher than the permissible levels for Cd and Pb, respectively. High Pb and Cd amounts in vegetables grown in an industrial area were also reported by Haque et al. [36] and by Gao et al. [28]. It should be underlined here that tomatoes accumulated the lowest Pb and Cd amounts, which in the case of Cd was also highlighted by Yang et al. [41]. The content of Cd in the analyzed vegetables was comparable to the results obtained by Gupta et al. [42], who determined from 0.14 to 0.23 mg·kg^−1^ of Cd for different vegetables. Moyo et al. [43] and Chowdhury et al. [35] found higher levels of Cd in tomatoes (0.6 mg·kg^−1^ and 0.62–2.88 mg·kg^−1^, respectively, for the cited authors) and in cabbage (0.54 mg·kg^−1^ and 0.91–3.52 mg·kg^−1^, respectively, for the cited authors) than in this study. It should be emphasized here that Cd and Pb are not essential elements for plant growth and can cause toxic effects in living organisms even at trace amounts [5,44]. 

The metal content in soils and plants provide general information concerning the current state of the environment, informing on their potential accumulation. However, when considering the negative impact of heavy metals on plants, especially on their edible parts, one should take into account their morphology and physiology, which will directly determine the individual intensity of heavy metal accumulation by a given plant [45]. Therefore, a more informative indicator allowing to reliably assess such a negative influence of metals and their accumulation intensity by plants seems to be provided by the bioconcentration factor (BCF). The bioconcentration factor (BCF) is widely used to assess toxicity of heavy metals as well as their translocation from soil to plants [11,46,47,48]. Additionally, Hu et al. [49] stated that the bioaccumulation values rather than the total contents of heavy metals should be taken into consideration, since more plants differ greatly in the degree of accumulation of heavy metals. According to Bounar et al. [50], a value of BCF < 0.1 indicates that vegetables are more successful at limiting the intake of heavy metals from contaminated soil. If values of BCF > 1, it shows that these plant species can be included in the category of heavy metal hyperaccumulators [51]. In this study, the range of BCF values, regardless of heavy metals and vegetables, amounted from 0.008 to 0.23, so they did not exceed 1. Despite differences between the minimum and maximum values of BCF, they depended on the vegetable and the zone in which the plant was grown. Regardless of the vegetable, the lowest values were recorded for zone III and the highest for zone II. Despite such differences, in most cases the values were statistically non-significant. The indicated differences in the BCF values for vegetables grown on soils from the II and III zones are the result of the amount of metals in the soil and in the plant. Generally, the higher the metal content found in the soil, the lower the BCF values were calculated for individual metals and vegetables. However, when interpreting this indicator, soil properties responsible for their sorption capacity, i.e., the potential binding of metals by soil colloids, should also be taken into account. Soils from zone II, compared to those from zone III, were characterized by lower CEC values and TC content (see Table 1). As indicated by Jakubus and Graczyk [10], the availability of metals for plants depends, among others, on such factors. Thus, it can be assumed that although in the soils from zone II, located further away from the emission source, the content of metals was lower, their bioavailability was higher, resulting in higher uptake of metals with crop yield and finally BCF value.

The lowest BCF values were determined for Pb (plant range: 0.008–0.01) and the highest for Cu (plant range: 0.05–0.23). Beetroot was characterized by the highest values of BCF (Cu: 0.09–0.23; Zn: 0.01–0.11; Cd: 0.03–0.013; Pb: 0.01–0.03), while cabbage had the smallest (Cu: 0.05–0.1; Zn: 0.01–0.05; Cd: 0.02–0.05; Pb: 0.008–0.02). The statement of Yang et al. [41] or Zhong et al. [52] that root plants have a high potential to accumulate heavy metals was proved by the outcomes of this study, because apart from beetroots high BCF values were also determined for carrots. This may be linked to the fact that the edible parts of these vegetables are in direct contact with contaminated soil. Although beetroot and carrot are classified as root plants, each plant belongs to a different botanical family (beetroot represents *Amaranthaceae* family and carrots *Umbelliferae*), which has its consequences in the different physiology and biochemistry of these plants. This will directly affect the different nutritional needs of plants, different resistance to the presence of heavy metals in the environment and the production of biomass yield. In general, greater biomass is associated with greater intake of nutrients, both essential and harmful. This was showed in the present study, because beetroot was characterized by a greater accumulation and concentration of heavy metals compared to carrots, which was also reflected in higher BCF values for this vegetable. At the same time, the theory by Gupta et al. [4] and Hu et al. [49] that leafy vegetables accumulate greater amounts of metals in comparison to non-leafy vegetables was not confirmed, because the lowest BCF values were obtained for cabbage being a leafy vegetable. Analysis of data from the literature indicates that generally the BCF values were considerably higher than in present study. Latif et al. [14] showed higher Zn BCF values from 6.13 to 12.9, Cd BCF in the range of 1–7.8 and Cu BCF from 4.33 to 14 for different vegetables. Also Moyo et al. [43] reported higher values of BCF Cd for cabbage. The results obtained by Ashraf et. al. [33] showed greater BCF Pb values from 0.10 to 0.08 for different vegetables. In turn, Chowdhury et al. [35] reported BCF Cd values for tomatoes grown in industrial regions in the range from 0.07 to 0.27, while for cabbage they amounted from 0.12 to 0.35. Heavy metals may enter the human body in different ways, but consumption of contaminated food is the main route [14,53]. The DIM value indicates the transfer of heavy metals from plants to humans. In this study, the daily intake of metals with vegetables was calculated for adults and the results show that generally consumption of the analyzed vegetables containing heavy metals poses practically no risk in the case of single cases, as they are less than 1. An exception was found for tomatoes cultivated in soils from all the analyzed zones. The values of DIM for Cu varied from 3.76 to 11.88 μg·kg^−1^ and for Zn from 7.27 to 18.08 μg·kg^−1^. Generally DIM values were significantly higher in zones III and IV rather than zones I and II. Haque et al. [36] compared DIM values for vegetables cultivated in industrial and non-industrial area and found considerably higher values in comparison to those calculated in this study. Regardless of the city zones, the calculated DIM values were the highest for Zn (range for plants: 0.17–18.08 μg·kg^−1^) and the lowest for Cd (range for plants: 0.001–0.12 μg·kg^−1^). Simultaneously for tomatoes the DIM values were the highest and for beetroots the lowest. Generally, the authors present significantly higher values of DIM calculated for vegetables. According to Latif et al. [14], the DIM of heavy metals for vegetables was 0.00166 (mg/day) for Cu, 0.0139 (mg/day) for Zn and 0.00012 (mg/day) for Cd. Higher DIM values for vegetables were shown in studies of Ashraf et al. [33], where DIM for tomatoes and cabbage for all metals amounted from 0.0002 to 0.0089 mg/day/kg weight. Data obtained by Mahmood and Malik [54] showed the DIM of heavy metals in the range from 0.0009 to 0.04 mg/day. Simultaneously the above-cited authors showed higher DIM values for beetroot or cabbage than for tomato and carrot. According to Latif et al. [14], PTDI amounted to 60 µg, 3 mg, 214 µg, and 60 mg per day for Cd, Cu, Pb, and Zn, respectively. Taking into account the above information and confronting it with the DIM values obtained in this study for metals, it can be stated that they were considerably lower than those given for PTDI. Therefore, it may be stated that the amounts of Cu, Zn, Cd, and Pb that could be theoretically consumed by adults with carrot, tomato, cabbage and beetroot products do not pose any health risk. 

There are many paths by which people get exposed to potential toxic metal(oid)s, and ingestion of vegetables contaminated with such elements could damage human health [55]. Considering the complexity of the negative impact of heavy metals on human health, it is reasonable to thoroughly assess the potential threat to humans from heavy metals consumed with food, especially as the digestive route is the most popular way for harmful elements to enter the human body. According to Jakubus and Bakinowska [11] HRI is an important tool to assess the negative impact of heavy metals on human health. According to US-EPA [56], HRI > 1.0 is considered hazardous to human health. In this study, HRI values were considerably higher for zones III and IV than I and II. The same pattern was noted regarding the content of heavy metals in soils and vegetables. Simultaneously, the values of HRI for Cu, Zn, Cd, and Pb in all the zones exceeded 1.0, which can indicate a potential significant unfavourable influence on human health. An exception was only observed in the case of beetroots cultivated in the urban zones (I, II), where HRI Zn ranged from 0.57 to 0.79 and for HRI Pb varied from 0.3 to 0.64 independently of the city zone. Regardless of plants, the highest values were calculated for Cu (2.52–296.94) and the lowest for Pb (0.3–30.65). Moreover, for beetroots, small values were calculated (range for plants: 0.3–7.57), while for tomatoes they were the biggest (11.98–296.94), indicating that the level of non-carcinogenic adverse health effect is alarmingly higher than the safe limit. Calculations show that the HRI for Cd in studied vegetables was also found to be significantly higher than standard limits. Thus, the HRI of Cd in tomatoes ranges from 38 to 123, from 5 to 14 in carrots, from 6 to 23 in cabbage, and from 1 to 4 in beetroots. Cd is a non-essential metal and highly toxic. Cd has shown chronic effects including lung cancer, respiratory diseases, and hair and bone problems [42]. Therefore, it is urgent to take appropriate measures to reduce the concentration of heavy metals in the environment in the study region. Consumption of unsafe concentrations of heavy metals continuously through food may lead to chronic accumulation of heavy metals in the human kidney and liver, consequently disrupting numerous biochemical processes and leading to cardiovascular, nerve, kidney and bone diseases [45]. Hu et al. [57], noted that excessive bioaccumulation of toxic heavy metals in vegetables may result in the unavailability of dietary nutrients to humans or cause health problems for both humans and the ecosystem. According to Li et al. [58], exposure to lead (Pb) may cause plumbism, anaemia, nephropathy, gastrointestinal colic, and central nervous system symptoms. High HRI values for tomato were also reported by Gao et al. [28]. The HRI values obtained in this study are significantly higher than those given in the literature by Mahmood and Malik [54], where HRI values for the metals were at the level of 0.02–0.45, and Latif et al. [14], who obtained HRI values < 1. Galal et al. [59] noted that the HRI values for Pb and Cd calculated for plants from contaminated sites were many times greater than 1.0 and consumption of such a plant product will pose a danger for the health of the local population. Growing vegetables on contaminated soils directly affects the quality of the products and this problem is closely related to food security. Despite the fact that urban gardening has not only aesthetic, but also economic attractiveness for the population of the city of Oskemen, the results of this study show significant negative aspects. In the light of obtained results the authors of this study do not recommend the use of vegetables grown in industrial areas due to the increased contents of heavy metals, as long-term consumption of contaminated vegetables can have adverse effects on human health. The results obtained indicate the need for further research, including the impact of contaminated vegetables on children as well as ongoing monitoring of the metal contents in the edible parts of cultivated plants in industrialized regions.

## 5. Conclusions

The excessive anthropogenic load in the city of Oskemen led to a deterioration in the environmental situation in general. In this study, in the soils of industrial zones the contents of the analyzed metals were found to be above the permissible limits. Generally, with a small exception for Cu and Zn in tomato, the analyzed metals exceeded thresholds given by FAO/WHO. Despite this fact, the obtained BCF values for the tested vegetables were very low, indicating low efficiency of heavy metal transfer from soil to plants, limiting their uptake. The DIM values of heavy metals for cabbage, beetroots and carrots in this study were estimated to be below 1, which indicates that the analyzed vegetables grown in the city area present little or no risk when consumed once. On the other hand, the DIM for Cu and Zn in tomatoes exceeded safe limits. It was found that consumption of vegetables grown in the zones of Oskemen could pose a significant non-carcinogenic risk to consumers, which was confirmed by the recorded HRI values. In the light of the presented results, it is reasonable to assess the usefulness of the parameters used in this study. In the authors’ opinion the best forecasting and estimation of negative effects of heavy metals on human health are provided by the use of HRI. Additionally, the outcomes of this study will help in the future planning of vegetable cultivation and the formulation of policies to strengthen human health by reducing and mitigating of the negative effect of heavy metals released into the environment as a result of anthropogenic activity. 

## Figures and Tables

**Figure 1 ijerph-20-00275-f001:**
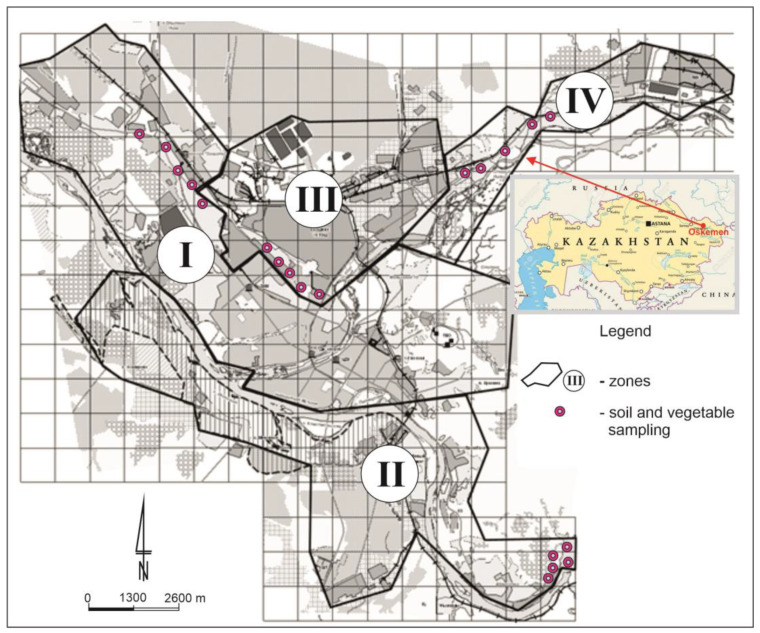
Study area with the location of zones and soil and vegetable sampling sites.

**Figure 2 ijerph-20-00275-f002:**
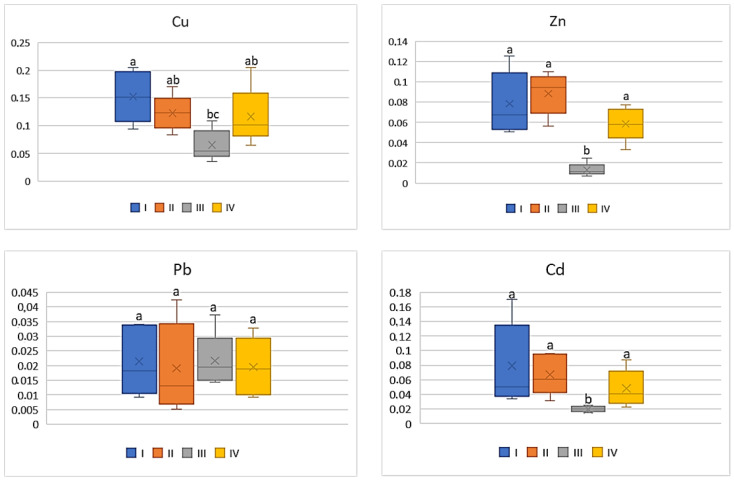
BCF values for carrots depending on the metal and study zones. The same letters mean no significant differences between the values.

**Figure 3 ijerph-20-00275-f003:**
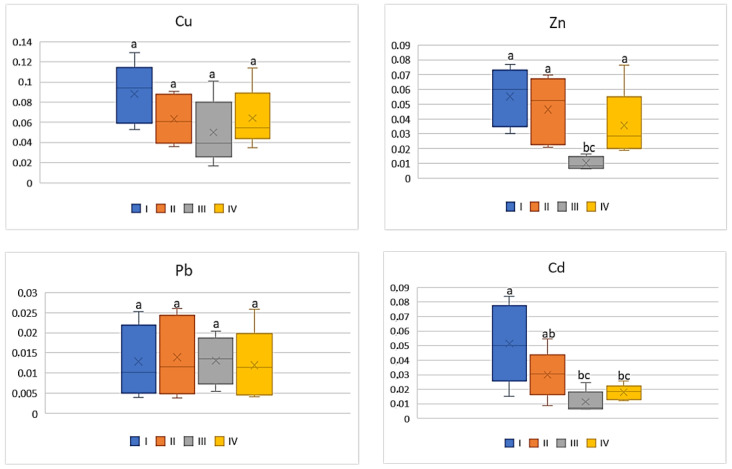
BCF values for tomato depending on the metal and study zones. The same letters mean no significant differences between the values.

**Figure 4 ijerph-20-00275-f004:**
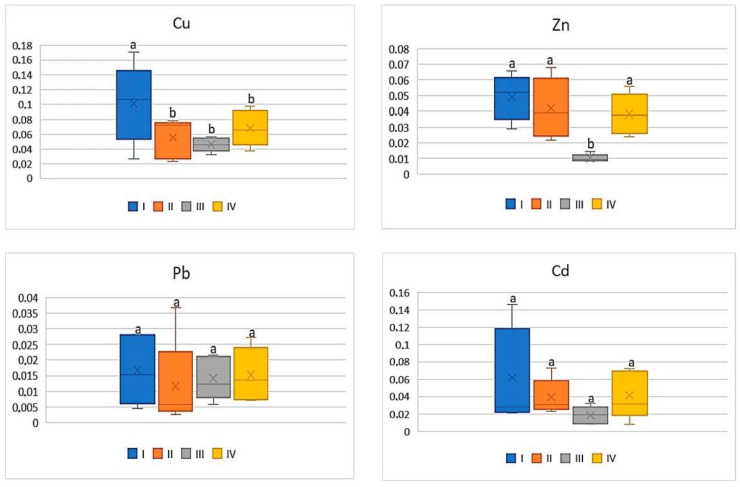
BCF values for cabbage depending on the metal and study zones. The same letters mean no significant differences between the values.

**Figure 5 ijerph-20-00275-f005:**
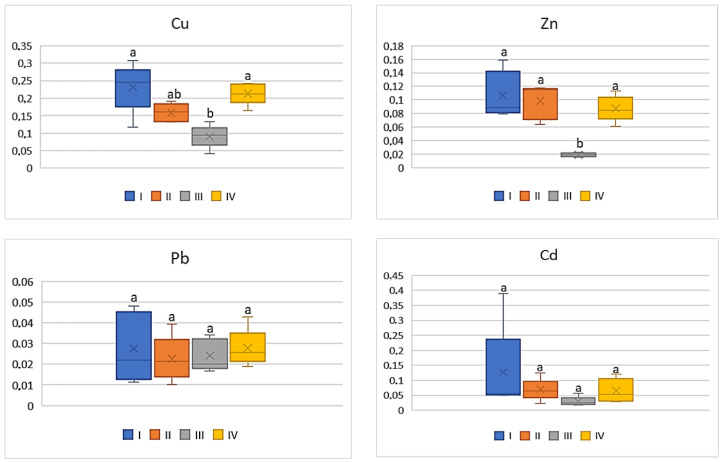
BCF values for beetroot depending on the metal and study zones. The same letters mean no significant differences between the values.

**Figure 6 ijerph-20-00275-f006:**
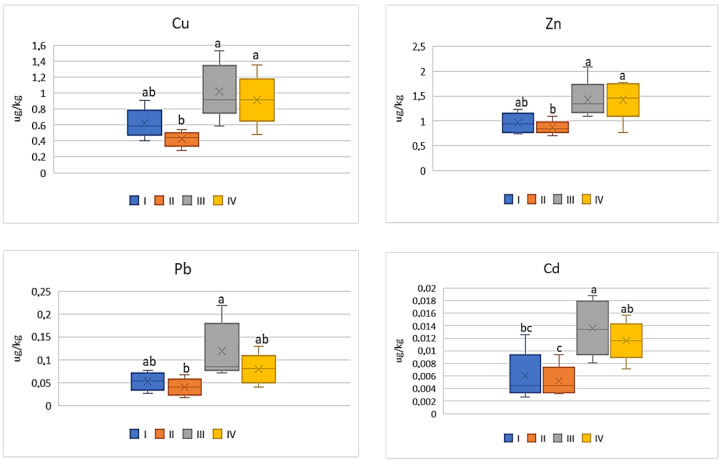
DIM values for carrot depending on the metal and study zones. The same letters mean no significant differences between the values.

**Figure 7 ijerph-20-00275-f007:**
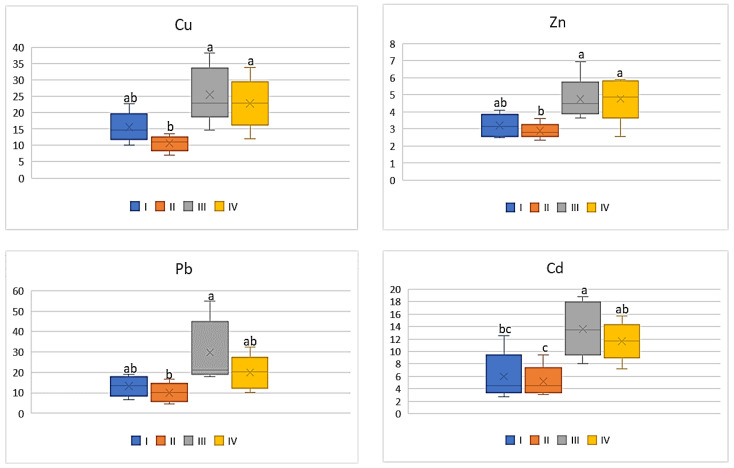
HRI values for carrot depending on the metal and study zones. The same letters mean no significant differences between the values.

**Figure 8 ijerph-20-00275-f008:**
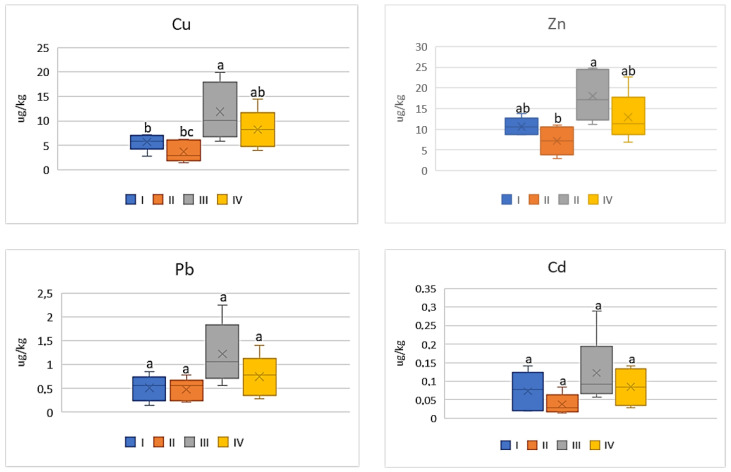
DIM values for tomato depending on the metal and study zone. The same letters mean no significant differences between the values.

**Figure 9 ijerph-20-00275-f009:**
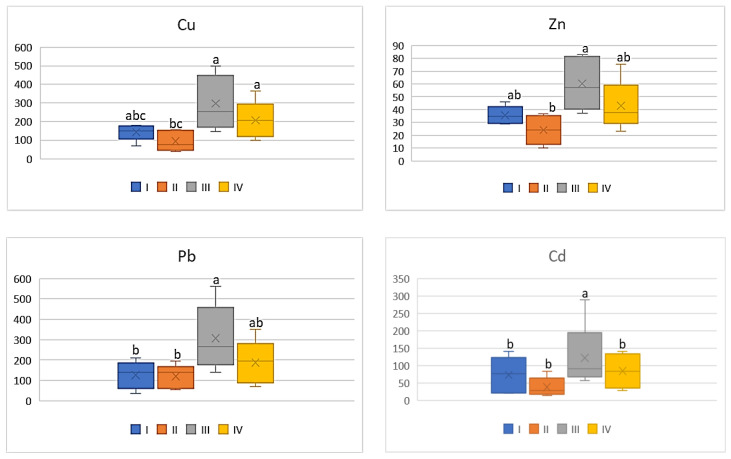
HRI values for tomato depending on the metal and study zones. The same letters mean no significant differences between the values.

**Figure 10 ijerph-20-00275-f010:**
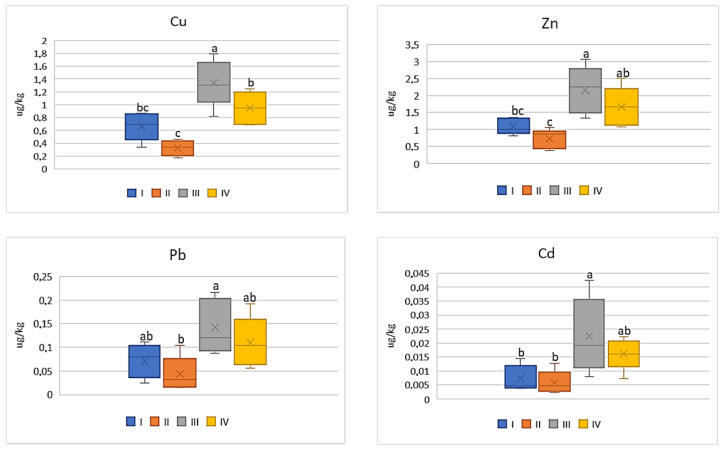
DIM values for cabbage depending on the metal and study zones. The same letters mean no significant differences between the values.

**Figure 11 ijerph-20-00275-f011:**
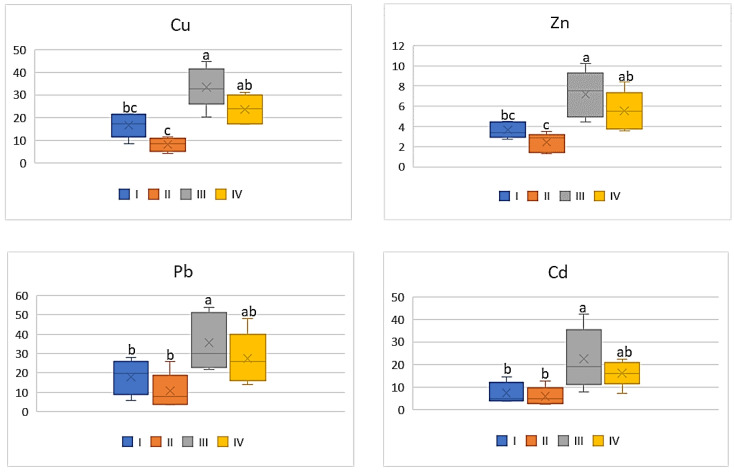
HRI values for cabbage depending on the metal and study zones. The same letters mean no significant differences between the values.

**Figure 12 ijerph-20-00275-f012:**
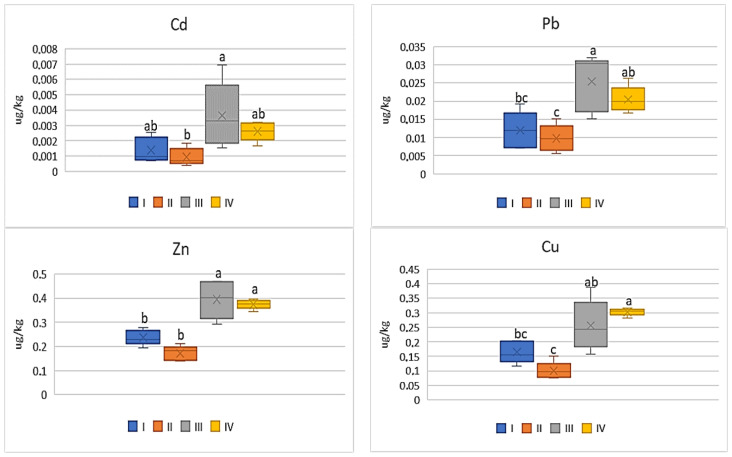
DIM values for beetroot depending on the metal and study zones. The same letters mean no significant differences between the values.

**Figure 13 ijerph-20-00275-f013:**
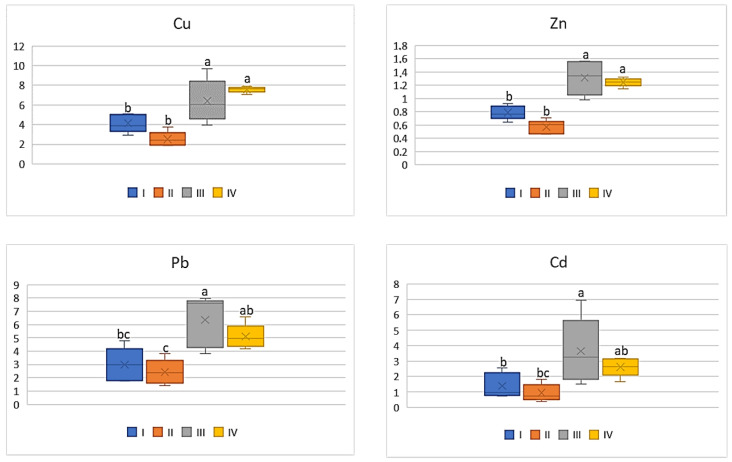
HRI values for beetroot depending on the metal and study zones. The same letters mean no significant differences between the values.

**Table 1 ijerph-20-00275-t001:** General characteristics of soils representing individual zones.

Zone	Distance from the Emission Centre (m)	% Clay	pH-CaCl_2_	CEC Cmol (+)·kg^−1^	TC %	TN %	C:N
I *	4500	3.8–6.1 ** 4.8 ± 0.9 ***	7.23–7.33 7.28 ± 0.05	12.8–17.0 14.7 ± 1.7	2.13–3.47 2.60 ± 0.63	0.15–0.24 0.20 ± 0.04	12.0–15.1 12.9 ± 1.2
II	13,000	4.1–5.3 4.4 ± 0.6	7.19–7.61 7.44 ± 0.18	8.1–16.5 12.8 ± 3.4	0.98–3.26 2.17 ± 0.91	0.09–0.27 0.17 ± 0.06	10.9–14.2 12.6 ± 1.2
III	2200	2.9–4.1 3.9 ± 0.7	6.42–7.54 7.40 ± 0.34	12.0–25.3 16.0 ± 5.4	2.12–5.55; 3.18 ± 1.38	0.17–0.36 0.23 ± 0.08	11.8–15.4 13.3 ± 1.5
IV	6000	3.9–5.1 4.5 ± 0.5	7.14–7.70 7.44 ± 0.26	10.6–18.4 14.7 ± 2.8	1.54–3.67 2.68 ± 0.78	0.12–0.26 0.20 ± 0.05	12.8–14.1 13.3 ± 0.6

* I—central urban zone, II—suburban zone, III—industrial zone, IV—north-eastern industrial zone ** range of values; *** mean values ± SD.

**Table 2 ijerph-20-00275-t002:** Metal content in soils and plants in relation to individual areas of the city (mg·kg^−1^).

Study Zones	Soil	Carrot	Tomato	Cabbage	Beetroot
	Cu				
I *	98.0 bc **	9.54 b	8.12 ab	8.30 bc	20.68 bc
II	79.38 c	9.54 b	5.34 b	4.10 c	16.62 c
III	366.2 a	22.92 a	16.86 a	16.78 a	37.90 a
IV	181.1 b	20.50 a	11.70 ab	11.86 ab	32.02 c
	Zn				
I	304.58 bc	21.5 ab	15.12 ab	13.72 bc	29.70 b
II	223.94 c	19.34 b	10.24 b	9.14 c	21.48 b
III	2645.13 a	31.88 a	25.66 a	27 a	49.28 a
IV	552.02 b	21.92 a	18.32 ab	20.82 ab	46.78 a
	Pb				
I	62.10 c	1.18 ab	0.72 a	0.90 ab	1.50 bc
II	58.66 d	0.90 bc	0.68 a	0.54 b	1.22 c
III	134.88 a	2.66 a	1.74 a	1.78 a	3.18 a
IV	58.66 b	1.78 ab	1.06 a	1.38 ab	2.56 ab
	Cd				
I	1.92 bc	0.13 bc	0.10 a	0.09 b	0.17 b
II	1.77 bc	0.12 c	0.05 a	0.07 b	0.12 b
III	15.2 a	0.31 a	0.17 a	0.28 a	0.46 a
IV	6.33 b	0.26 ab	0.12 a	0.20 ab	0.33 a

* I—central urban zone, II—suburban zone, III—industrial zone, IV—north-eastern industrial zone. ** The same letters mean no significant differences between the values.

## Data Availability

Not applicable.

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
