# Peer review of "Health Risk of Heavy Metals Related to Consumption of Vegetables in Areas of Industrial Impact in the Republic of Kazakhstan—Case Study for Oskemen"

_ijerph, 2022, doi:10.3390/ijerph20010275_

Round 1

Reviewer 1 Report

General comments

1.     The science behind the observed BCF values highlighted on page 6 needs to be explained in the discussion.

2.     In Figures 2, 3, and 4 onwards, no meaning has been given to the letters a, ab, bc, etc.

3.     Since the HRI(s) have been deemed most vital in assessing the potential magnitude of toxicity in humans, it needs to be expounded more on.

4.     Science behind the high BCF in zone one and low BCF in zone three needs to be highlighted as the information on the disparity between the two is vague.

5.     Why does beetroot have a higher hyperaccumulation of heavy metal ions than carrot and they are both root tubers? The authors have just mentioned so and not much detail has been directed as to the ‘why’.

Reviewer 2 Report

Dear Authors,

Your research work is interesting because it shows the health risk of heavy metals associated with consuming vegetables in industrially impacted areas. The research covered the industrial city of Oskemen in the Republic of Kazakhstan. This paper is prepared in the usual way for scientific work. Manuscript is prepared carefully. However, I have a few comments. Some are debatable:

Line 12: You write, “Among various heavy metal sources the metallurgic industry is the most threatening”. On what basis do you say that? Please explain it.

Line 144. “zone**” – please add space.

Line 136 and Table 1 are "N tot", but line 145 is TN. Please unify the record.

Section 2.2. Add the country of manufacture of the apparatus used.

Line 206. “Table2” – please add space.

Line 208. “indus- trial” – please connect word.

Line 334. “[24,25]” – please add space.

Line 387. Please remove "(2021)"

Lines 429 and 434. Red font. Please correct it.

In my opinion, the results, discussion and conclusion chapters are well written. However, explain to what extent the tested soils were contaminated by HMs. Are there background studies for these HMs in the Republic of Kazakhstan. Or are there classes (categories) of pollution?

The language appears to be correct, but I don't feel qualified to judge about the English language and style.

I recommend for publication in IJERPH after the indicated corrections.

Good luck!

Sincerely yours

Reviewer
